# Antiplasmodial, Trypanocidal, and Genotoxicity *In Vitro* Assessment of New Hybrid α,α-Difluorophenylacetamide-statin Derivatives

**DOI:** 10.3390/ph16060782

**Published:** 2023-05-24

**Authors:** Carlos Fernando Araujo-Lima, Rita de Cassia Castro Carvalho, Sandra Loureiro Rosario, Debora Inacio Leite, Anna Caroline Campos Aguiar, Lizandra Vitoria de Souza Santos, Julianna Siciliano de Araujo, Kelly Salomão, Carlos Roland Kaiser, Antoniana Ursine Krettli, Monica Macedo Bastos, Claudia Alessandra Fortes Aiub, Maria de Nazaré Correia Soeiro, Nubia Boechat, Israel Felzenszwalb

**Affiliations:** 1Laboratório de Biologia Celular, LBC Instituto Oswaldo Cruz—FIOCRUZ, Rio de Janeiro 21041-250, RJ, Brazil; biomed.carlos@gmail.com (C.F.A.-L.); ju.siciliano@gmail.com (J.S.d.A.); ks@ioc.fiocruz.br (K.S.); 2Laboratório de Mutagênese Ambiental, LabMut Instituto de Biologia Roberto Alcantara Gomes, IBRAG—UERJ, Rio de Janeiro 22050-020, RJ, Brazil; lizabiomed@gmail.com; 3Programa de Pós-Graduação em Biologia Molecular e Celular, Instituto Biomédico—UNIRIO, Rio de Janeiro 20211-030, RJ, Brazil; aiub.claudia@gmail.com; 4Departamento de Síntese de Fármacos, Instituto de Tecnologia em Fármacos, Farmanguinhos—FIOCRUZ, Rua Sizenando Nabuco 100, Manguinhos, Rio de Janeiro 21041-250, RJ, Brazil; ritacarvalho22@gmail.com (R.d.C.C.C.); riorosar@gmail.com (S.L.R.); debora.marinho@fiocruz.br (D.I.L.); mmacedobastos@gmail.com (M.M.B.); 5Programa de Pós-Graduação em Química, PGQu, Instituto de Química, Universidade Federal do Rio de Janeiro, Rio de Janeiro 21941-853, RJ, Brazil; kaiser@iq.ufrj.br; 6Programa de Pós-Graduação em Farmacologia e Química Medicinal, ICB-UFRJ, Rio de Janeiro 21941-902, RJ, Brazil; 7Laboratório de Malária, Centro de Pesquisas René Rachou, CPqRR—FIOCRUZ, Belo Horizonte 30190-002, MG, Brazil; carolcaguiar@yahoo.com.br (A.C.C.A.); antoniana.krettli@fiocruz.br (A.U.K.)

**Keywords:** phenylacetamides, atorvastatin, hybrid compounds, chemical synthesis, drug development, antiparasitic activity, genotoxicity assessment

## Abstract

Background: Statins present a plethora of pleiotropic effects including anti-inflammatory and antimicrobial responses. A,α-difluorophenylacetamides, analogs of diclofenac, are potent pre-clinical anti-inflammatory non-steroidal drugs. Molecular hybridization based on the combination of pharmacophoric moieties has emerged as a strategy for the development of new candidates aiming to obtain multitarget ligands. Methods: Considering the anti-inflammatory activity of phenylacetamides and the potential microbicidal action of statins against obligate intracellular parasites, the objective of this work was to synthesize eight new hybrid compounds of α,α-difluorophenylacetamides with the moiety of statins and assess their phenotypic activity against *in vitro* models of *Plasmodium falciparum* and *Trypanosoma cruzi* infection besides exploring their genotoxicity safety profile. Results: None of the sodium salt compounds presented antiparasitic activity and two acetated compounds displayed mild anti-*P. falciparum* effect. Against *T. cruzi*, the acetate halogenated hybrids showed moderate effect against both parasite forms relevant for human infection. Despite the considerable trypanosomicidal activity, the brominated compound revealed a genotoxic profile impairing future *in vivo* testing. Conclusions: However, the chlorinated derivative was the most promising compound with chemical and biological profitable characteristics, without presenting genotoxicity *in vitro*, being eligible for further *in vivo* experiments.

## 1. Introduction

Statins are competitive inhibitors of hydroxy-3-methylglutaryl coenzyme A (HMG-CoA) reductase, an enzyme that catalyzes the NADPH-dependent reduction of HMG-CoA to mevalonate in the cholesterol and ergosterol biosynthesis (Figure 1). Besides their role in cholesterol control, statins present pleiotropic effects including acting on inflammatory responses and antimicrobial effects [1,2,3]. Some of them, such as atorvastatin (AVA, **1**, Figure 2), which is a lipid-lowering agent acting on HMG-CoA reductase present promising antiparasitic activity against intracellular protozoan parasites such as *Toxoplasma gondii* [4], *Trypanosoma cruzi* [5] and *Plasmodium falciparum* [6]. However, the synthesis of less toxic statins and a deeper understanding of the mechanisms underlying statin interplay may contribute to minimizing drug interactions and reducing their adverse side effects such as skeletal muscle toxicity [7].

Chagas disease (CD), caused by the protozoan *Trypanosoma cruzi*, is an important public health problem, affecting more than 6 million people, resulting in ≃10,000 annual deaths and 528,000 Disability-adjusted life years (DALYs) [8,9,10]. The available therapeutic options for CD are limited to benznidazole (Bz) (**2**) and nifurtimox (**3**) (Figure 2). Both display relevant limitations including severe side effects and lack of efficacy in the later chronic phase, besides the occurrence of naturally resistant strains [11,12]. Lovastatin (**4**) (Figure 2), able to competitively inhibit the HMG-CoA of *T. cruzi* epimastigotes [13], is primarily localized in the parasite mitochondrion [14]. Lovastatin in combination with ketoconazole or terbinafine was active *in vivo* against *T. cruzi* infection [15]. Simvastatin (**5**) (Figure 2) improved the cardiac remodeling in *T. cruzi*-infected dogs [16] but was unable to reduce parasitemia levels and cardiac parasite load in a mouse model of this parasite infection [17].

Malaria is another parasitic disease considered a huge public health problem in more than 95 countries, resulting in 247 million cases in 2021, 619,000 annual deaths, 3.2 billion people at risk and about 55,111,095 DALYs [18]. The disease is caused by protozoa of the genus *Plasmodium* and the life cycle in mammalians occurs under different stages infecting the liver and erythrocytes. In endemic areas such as on the African continent, the deaths are mainly related to cerebral malaria, which is the most severe manifestation caused by *Plasmodium falciparum*. The socio-economic development of the affected regions is highly impaired, corroborating the importance of prevention and treatment of this pathology [14]. Thus, due to the well-known pleiotropic effects such as neuroprotective and anti-inflammatory activity, AVA has been tested as an adjuvant in the treatment of experimental malaria [19,20] and its cerebral models [21]. Owing to the increased ability of *Plasmodium* to acquire resistance to chemotherapeutic agents associated with well-known toxicity aspects of the current antimalarial drugs [14], the discovery and development of alternatives for this severe pathology is an urgent need. Up to now, studies fail to demonstrate the cross-resistance *in vitro* between AVA and antimalarials, suggesting the occurrence of different modes of action, which is a potential prediction that AVA may be a promising alternative for malaria therapy [22,23,24,25]. In this context, our group synthesized pyrrolic hybrids of AVA with aminoquinolines and assayed against *P. falciparum* and the finding demonstrated an improved activity as compared to chloroquine (**6**) and also being less toxic than primaquine (**7**) (Figure 2) [26].

Hybrid molecules represent a very promising and current approach for antiparasitic drug development. Molecular hybridization based on the combination of pharmacophoric moieties has emerged as an important strategy for the development of new drugs that are able to act as multitarget ligands [26]. In this sense, anti-inflammatory non-steroidal drugs (NSAIDs such as diclofenac (**8**) Figure 3) have been used as adjuvant treatment in patients with malaria [27] and α,α-difluorophenylacetamides **9** (Figure 3) are potent pre-clinical NSAID analogs of diclofenac [28,29]. Both **8** and **9** have chemical structures based on phenylacetic acid and phenylacetamide, respectively. Phenylacetamides also display a large spectrum of biological activity [30,31].

Finally, another fundamental initial step in the drug discovery pipeline is to assess the potential toxicological profile of a new hit as recommended by OECD, performing at least two *in vitro* toxicological assays before moving to *in vivo* experimentation [32]. Usually, the first set of analyses regarding toxicological evaluation is related to the *Salmonella*/Microsome reverse mutation assay that allows observing point mutations in DNA sequence, following to evaluation of clastogenicity potential, through micronucleus assay [33].

Having in account the anti-inflammatory activity of phenylacetamides and also the potential microbicidal action of statins against *P. falciparum* and *T. cruzi*, the present aim was synthesize new hybrid compounds of α,α-difluorophenylacetamides with the moiety of statins (3*S*,5*S*)-3,5-dihydroxyheptanoic **13** (**a**–**d**) and **14** (**a**–**d**) as represented in the planning in Figure 3 and perform phenotypic screenings against *in vitro* models of these parasitic infections besides exploring their potential genotoxicity safety profiles.

## 2. Results and Discussion

### 2.1. Chemistry

The synthetic route (Figure 4) starts with hydrogenation of the commercially available nitrile compound **10**, which is an intermediate of synthesis of AVA, in Parr reactor, using Raney nickel or palladium on carbon 10% as a catalyst to produce the amino intermediate **11** [34,35]. This amino intermediate **11** is used in a nucleophilic addition on N-acetyl-3,3-difluoro-2-oxoindoles **12** (**a**–**d**) to furnish the protected α,α-difluorophenylacetamides **13** (**a**–**d**) [28]. N-acetyl-3,3-difluoro-2-oxindoles **12** (**a**–**d**) were obtained from fluorination with DAST (diethylamino sulphortrifluoride) of the corresponding N-acethylisatins **16** (**a**–**d**), which were obtained by acetylation of isatins **15** (**a**–**d**).

The products **14** (**a**–**d**) were obtained by hydrolysis under acidic conditions and transformed into salt with aqueous NaOH [36]. All compounds **13** (**a**–**d**) and **14** (**a**–**d**) were fully characterized by FTIR, NMR ^1^H and ^13^C, proving the formation of the desired products.

### 2.2. Biological Evaluation

The results of the antiplasmodial activity and cytotoxicity on HepG2 of the new compounds are summarized in Table 1. Compounds **13b**, **13c** and AVA were active against *P. falciparum*, with EC_50_ values of 14.26 μM, 11.78 μM and 10.30 μM, respectively. The preliminary cytotoxicity finding demonstrated a lack of toxic events up to 400 μM. Among the tested compound, **13b** and **13c** were the best exhibiting similar potency as AVA and can be considered as future prototypes for the development of new antimalarial drugs.

Table 2 shows the effects of **13** (**a**–**d**) and **14** (**a**–**d**) against bloodstream trypomastigotes of *T. cruzi* Y strain (moderately resistant to nitroderivatives as Bz) and mouse cardiac cell cultures (CC). It is possible to observe that **13c** and **13d** presented mild activity on the parasite, with EC_50_ corresponding to 28.20 µM and 23.18 µM, respectively, while AVA was slightly more potent than Bz (EC_50_ 7.07 µM and 13 µM, respectively) (Table 2). All the tested compounds **13** (**a**–**d**) and **14** (**a**–**d**) had LC_50_ higher than 500 µM after 24 h of exposure to the CC.

The inactivity of the salts **14** (**a**–**d**) could be attributed to their instability and thus only the **13** (**a**–**d**) were moved to further *in vitro* analysis to inspect their effect against intracellular forms of *T. cruzi* (Table 3) and their potential toxicological profile *in vitro* (Table 4, Figure 3 and Figure 4).

Using as a first filter, a fixed concentration of 10 µM (corresponding to the EC_90_ value of Bz against intracellular forms of *T.cruzi*, see [37]) only **13c** and **13d** induced a considerable antiparasitic effect (≥50% of reduction on the intracellular parasite load), being both more active than AVA (Table 3). Next, further analysis using increasing concentrations (up to 50 µM) confirmed the promising effect of both derivatives presenting EC_50_ values of 9.24 µM and 11.42 µM for **13c** and **13d**, leading to SIs of 16.7 and 10.0, respectively. On the other hand, **13a** and **13b** were less effective against intracellular forms of *T. cruzi*, exhibiting EC_50_ ≥50 µM (Table 3).

Regarding the trypanocidal activity of the studied statins, only the halogenated derivatives presented potential trypanosomicidal effects. **13c** and **13d** showed a similar range of activity, toxicity, and selectivity against the two different discrete typing units (DTU) of *T. cruzi* used in these phenotypic screenings. Against the intracellular forms of the Tulahuen strain (DTU VI), the chlorinated compound **13c** was more selective than the brominated **13d** although no major differences could be noticed when bloodstream trypomastigotes were assayed using the Y strain (DTU II). Our data corroborate previous *in vitro* studies that demonstrated the efficacy of lovastatin and simvastatin against epimastigote forms of the Y strain [16,38].

Next, as the risk assessment by Ames Test is mandatory for potential chemical entities for novel pharmaceutical clinical arsenal [33], this analysis was conducted with all the studied compounds (Table 4). Compound **13a** presented cytotoxicity to TA102 and TA104 exhibiting LC_50_ in the absence and in the presence of metabolic activation, corresponding to 1478.7 ± 56.1 µM 1280.0 ± 21.1 µM, respectively.

**13c** presented toxic concentrations to TA104 in the presence of metabolic conditions (LC_50_ = 1390.0 ± 14.8 µM). **13b** showed LC_50_ >1500 µM for all tested strains both in the absence and presence of the S9 mix. **13d** was the most active against the tested bacteria, being toxic to TA97 (LC_50_ = 1490.0 ± 36.5 µM), TA98 (LC_50_ = 1260.0 ± 46.8 µM), TA100 (LC_50_ = 120.5 ± 14.1 µM) and TA102 (LC_50_ = 14.7 ± 1.8 µM) in absence of S9 mix and toxic to TA100 (LC_50_ = 1450.0 ± 18.2 µM) and TA102 (LC_50_ = 442.8 ± 32.1 µM) after exogenous metabolic activation. Besides the cytotoxicity, **13a** and **13d** were capable to induce significant mutagenicity to TA98 (−S9) and TA97 (−S9/+S9), respectively, at concentrations above 300 µM. Despite cytotoxicity, both **13b** and **13c** were not mutagenic at all of the described experimental conditions (Table 4).

Finally, the mutagenic profile was checked using a micronuclei approach employing both CHO-K1 and HepG2 cell lineages as previously validated by [39] to detect *in vitro* genotoxicity. These authors classified the performance scores of these cell lineages for *in vivo* genotoxicity and found a high translation of both models at the beginning of drug discovery steps, representing a useful tool to assess genotoxic potential at the earlier stages. Our data conducted in parallel to WST-1 colorimetric assay to assess cytotoxicity showed that all the derivatives (**13a**–**d**) presented LC_50_ >1000 µM after exposures of 3 h and 24 h with CHO-K1 or HepG2 cell lineage. In micronuclei assay using CHO-K1 (Figure 5), after 3 h of exposure, **13a** induced a significant increase in micronuclei formation at 200 µM and presented a significant concentration-response decrease in cellular viability, being considered cytotoxic (Figure 5A). **13b** and **13c** (Figure 5B,C) were non-cytotoxic, non-clastogenic and did not affect the mitotic index of the ovarian cells. **13d** (Figure 5D) was significantly cytotoxic to ovary cell culture in all tested concentrations and genotoxic upon 1000 µM. *In vitro* micronuclei assay in HepG2 showed that the compounds **13a** (Figure 6A) and **13c** (Figure 6C) did not induce micronuclei formation in HepG2 cells, after 3 h of exposure, but just **13a** and AVA presented significant cytotoxic effects upon 1000 µM. The compound **13b** (Figure 6B) had a statistical enhancement, in comparison to the control, in genotoxicity and cytotoxicity parameters to hepatic cells at 2000 µM and **13d** (Figure 6D), besides it was non-cytotoxic, induced a significant micronuclei formation in HepG2 cells after the exposure to the three tested concentrations.

Our finding using statin derivatives demonstrated a similar profile of AVA being non-mutagenic neither genotoxic after *in vitro* nor *in vivo* toxicological screening [40]. The derivative **13c** did not present a genotoxic response and was only cytotoxic on TA104 in the presence of metabolic activation, presenting an LC_50_ >1200 µM (Figure 5 and Figure 6). **13b** had no mutagenic potential in our bacterial model and only increased micronucleation in HepG2 cells at >2000 µM. **13b** genotoxic effect can be related to the aromatic ring metabolism in the terminal region. The presence of a toluyl radical, after the oxidative process by CYP enzymes, can produce epoxide radicals capable to produce DNA adducts, inducing its genotoxicity [41].

The compound **13a** enhanced the number of revertants for TA98 in the absence of an exogenous metabolic condition, which suggests direct mutagen-promoting frameshift mutations by insertion or deletion of G:C base pairs in *hisD3052* hotspot genic locus [42]. On the other hand, after metabolic activation, its mutagenic response was not observed and, probably, CYP enzymes can inactivate the **13a** mutagenic effect. The evidence that **13a** causes DNA damage directly and loses its genotoxic potential after metabolization are reinforced by the data on eukaryotic cells since it was genotoxic and cytotoxic to CHO-K1 cells and non-genotoxic to HepG2 cells even though cytotoxicity remains present.

The presence of bromine on the aromatic ring of **13d** can explain its cytotoxic and genotoxic effects [43]. This compound was mutagenic to TA97 both in the absence and presence of exogenous metabolization, acting as a frameshift mutagen, causing the addition of G:C base pairs in *hisD6610* locus. Notwithstanding, **13d** exposure was induced genotoxicity in ovarian and hepatic cell eukaryotic cultures (Figure 5 and Figure 6). The presence of bromine in organic molecules is related to its genotoxic potential. Tsuboy and coworkers (2007) [44] detected the genotoxic, mutagenic, and cytotoxic effects of a commercial dye that presents a bromide in a phenyl-amine ring.

The cytotoxic effects in the bacterial model and the trypanocidal activity presented by **13c** and **13d** can be related to the presence of halogens. Previous studies evaluating some new halogen-containing 1,2,4-triazolo-1,3,4-thiadiazines demonstrated that the chlorinated and brominated compounds were effective as antimicrobials against both Gram-positive and -negative bacteria [45].

An anti- *T. cruzi* hit compound must reach EC_50_ ≤10 µM against intracellular forms of representative strains relevant for mammalian infection (presently evaluated such as Y and Tulahuen strains belonging to TcII or TcIV discrete typing units, respectively) [11,46]. Additionally, hit compounds should not present structural alerts of genotoxicity at least performed by in silico platforms. According to the literature, to move a compound from hit to lead classification it is necessary to reach at least a 10–20-fold increase in potency and selectivity without presenting signals of *in vitro* mutagenicity or genotoxicity evidence [11]. According to these, the most promising derivative presently evaluated against *T.cruzi* are **13c** and **13d** which displayed activity against both parasite forms and upon different strains belonging to distinct DTUs. However, due to the genotoxicity profile of **13d**, prototype **13c** merits chemical optimization aiming to identify a novel lead for CD therapy. Our group has contributed to the comprehension of AVA efficacy concerning *T. cruzi* infection [5] and the particular role of pentapryrrolic moiety of AVA in its trypanocidal activity, hence another series of AVA-aminoquinolinic hybrid compounds which were highly effective, less toxic and more selective than AVA and Bz [47]. Regarding antiplasmodial activity, compounds **13b**, **13c** were active against *P. falciparum*, with EC_50_ values <14 μM with lack of toxic events up to 400 μM and can also be considered as future prototypes for the development of new antimalarial drugs. In this sense, the optimization of the statin hybrid derivatives is largely recommended to identify novel alternatives for neglected diseases and justify further studies *in vitro* and *in vivo*. In Table 5, the summarized outcomes of each newly synthesized compound were detailed.

## 3. Conclusions

Eight new hybrid compounds of α,α-difluorophenylacetamides with the moiety of statins (3*S*,5*S*)-3,5-dihydroxyheptanoic **13** (**a**–**d**) and **14** (**a**–**d**) were synthesized and their biological aspects evaluated. None of the sodium salt form compounds (**14a**–**d**) presented antiparasitic activity and two acetate form compounds (**13b** and **13c**) presented mild anti-*P. falciparum* activity when compared with AVA and the reference drug (chloroquine). Against *T. cruzi*, **13c** and **13d** presented moderate effects against both parasite forms relevant for human infection although less potent than Bz. However, despite the considerable trypanocidal activity, compound **13d** displayed a genotoxic profile that turns it unfeasible for moving it to *in vivo* testing. The derivative **13c** was the only which presents promising chemical and biological characteristics as it does not also present genotoxicity *in vitro* and has a quite considerable selectivity (>17) and thus justifying further future *in vivo* experiments as the next step of the CD experimental chemotherapy flowchart.

## 4. Materials and Methods

### 4.1. Chemistry

All reagents and solvents used were analytical grade. Thin layer chromatography (TLC) was performed using a Merck TLC Silica gel 60 F254 aluminum sheets 20 × 20 cm (eluent chloroform/methanol 9:1). The melting points (m.p.) were determined using a Büchi model B-545 apparatus. The ^1^H, ^13^C and ^19^F nuclear magnetic resonance (NMR) spectra were generated at 400.00, 100.00 and 376.00 MHz, respectively, in a BRUKER Avance instrument equipped with a 5 mm probe. Tetramethylsilane was used as an internal standard. The chemical shifts (δ) are reported in ppm, and the coupling constants (J) are reported in Hertz. The Fourier transform infrared (FT-IR) absorption spectra were recorded on a Shimadzu mode IR Prestige-21 spectrophotometer through KBr reflectance to **13** (**a**–**d**) samples and attenuated total reflection technique (ATR) for **14** (**a**–**d**) samples. Mass spectrometry with electrospray ionization in positive mode [ESI-MS (+)] was carried out in Waters ^®^ Micromass ZQ4000 equipment (Milford, MA, USA). Values are expressed as mass/charge ratio (*m*/*z*) and are equivalent to the molecular mass of the substance plus a proton. The HRMS data were obtained using LC–MS Bruker Daltonics MicroTOF (Yokohama, Kanagawa, Japan) (analyzer time of flight). The analysis by high-performance liquid chromatography (HPLC) was performed on Shimadzu liquid chromatograph using Supelcosil LC-8 column (Kyoto, Japan) (250 mm × 4.6 mm × 3 μm) and as mobile phase acetonitrile: potassium phosphate buffer 0.01 mol/L, pH 5.8, flow 1 mL/min.

### 4.2. Synthesis

#### 4.2.1. Preparation of *tert*-Butyl 2-((4R,6R)-6-(2-Aminoethyl)-2,2-dimethyl-1,3-dioxan-4-yl)acetate (11)

In a Parr reactor vessel was added intermediate 1,3-dioxane-4-acetic acid, 6-(cyanomethyl)-2,2-dimethyl-1,1-dimethylethyl ester **10** (10.3882 g, 0.0386 mol), methanol (110 mL), Raney-Ni catalyst (8.07 g, 0.1375 mol) or Pd © 10% (1 g) and 15–18% NH_4_OH solution (8 mL). The reaction was kept for 48 h under stirring and a hydrogen atmosphere (50 psi/3.4 atm). Then the reaction was filtered, concentrated, washed with methanol and concentrated again, giving an oil product. Yield: 90%. ESI-MS: *m*/*z* 274.0 [M + H]^+^. MSGC (*m*/*z*): 200 (70%); 158 (54%); 142 (100%); 100 (81%); 72 (74%). ^1^H NMR (400 MHz, MeOD) δ: 1.17–1.26 (1H, m, CHCH_2_CH_2_); 1.33 (3H, s, C(CH_3_)_2_); 1.45 (9H, s, C(CH_3_)_3_); 1.48 (3H, s, C(CH_3_)_2_); 1.61; 1.64 (1H, dt, *J* = 2; 12 Hz; CHCH_2_CH_2_); 1.74–1.93 (2H, m, CHCH_2_CH); 2.27–2.42 (2H, m, CHCH_2_Ac); 3.04–3.20 (2H, m, CH_2_NH); 4.06–4.13 (1H, m, CH); 4.30–4.36 (1H, m, CH). ^13^C NMR (100 MHz, MeOD) δ: 20.22; 28.46; 30.46; 33.68; 37.10; 43.63; 46.13; 67.50; 68.56; 81.99; 100.39; 172.16. IR (ATR, cm^−1^): 3408; 2979; 2944; 2732; 1719; 1366; 1258; 1154; 943.

#### 4.2.2. General Preparation of *tert*-Butyl 2-((4*R*,6*R*)-6-(2-(2-(2-Acetamido-5-subtitutedphenyl)-2,2-difluoroacetamido)ethyl)-2,2-dimethyl-1,3-dioxan-4-yl)acetate **13** (**a**–**d**)

In a flask was added amine **11** (0.2753 g, 0.001 mol), 5-substituted-1-acetyl-3,3-difluoroindolin-2-ones **12** (**a**–**d**) (0.001 mol), prepared according to the methodology previously described [29], and anhydrous DMF (5 mL) under magnetic stirring. The reaction mixture was stirred at 80 °C for 24 h. DMF solvent was removed in a rotary evaporator and pure products **13** (**a**–**d**) were precipitated in pure form with 25–70% yields with ice-cold water.

*tert*-Butyl 2-((4*R*,6*R*)-6-(2-(2-(2-acetamidophenyl)-2,2-difluoroacetamido)ethyl)-2,2-dimethyl-1,3-dioxan-4-yl)acetate (**13a**).

Yellow solid. Yield: 70%. m.p. 93–96 °C. HRMS (ESI) calc. for C_24_H_34_F_2_N_2_O_6_ 484.2385, found [M+1]^+^ 484.2385. HPLC grade: 96%. ^1^H NMR (400 MHz, CD_3_OD) δ: 1.12 (1H, q, *J* = 12 Hz, CHCH_2_CH); 1.28 (3H, s, C(CH_3_)_2_); 1.34 (3H, s, C(CH_3_)_2_); 1.44 (9H, s, C(CH_3_)_3_); 1.51 (1H, dt, *J* = 2; 12 Hz, CHCH_2_CH); 1.59–1.70 (2H, m, CHCH_2_CH_2_); 2.16 (3H, s, NHCOCH_3_); 2.24–2.35 (2H, m, CHCH_2_Ac); 3.38–3.44 (2H, m, CH_2_NH); 3.86–3.90 (1H, m, CH); 4.19–4.24 (1H, m, CH); 7.30 (1H, t, *J* = 6 Hz, H5); 7.51 (1H, t, *J* = 6 Hz, H4); 7.61 (1H, d, *J* = 6 Hz, H6); 7.86 (1H, d, *J* = 6 Hz, H3). ^13^C NMR (100 MHz, CD_3_OD) δ: 19.85; 23.95; 28.35; 30.36; 36.11; 37.26; 37.40; 43.59; 67.55; 68.37; 81.77; 100.02; 115.76 (t, *J* = 253 Hz); 126.37; 126.65–127.04 (m); 127.19; 132.71; 137.03 (t, *J* = 3 Hz); 166.62 (t, *J* = 31 Hz); 171.60; 172.03. ^19^F (376 MHz, CD_3_OD) δ: −104.50. IR (KBr, cm^−1^): 3307; 3240; 2983; 2941; 2862; 1728; 1699; 1674; 1554; 1369; 1257; 1147; 1087.

*tert*-Butyl 2-((4*R*,6*R*)-6-(2-(2-(2-acetamido-5-methylphenyl)-2,2-difluoroacetamido)ethyl)-2,2-dimethyl-1,3-dioxan-4-yl)acetate (**13b**)

Ivory-white solid. Yield: 44%. m.p. 116–119 °C. HRMS (ESI) calc. for C_25_H_36_F_2_N_2_O_6_ 498.2541, found [M + 1]^+^ 498.2541. HPLC grade: 94%. ^1^H NMR (400 MHz, CD_3_OD) δ: 1.12 (1H, q, *J* = 12 Hz, CHCH_2_CH); 1.26 (3H, s, C(CH_3_)_2_); 1.33 (3H, s, C(CH_3_)_2_); 1.44 (9H, s, C(CH_3_)_3_); 1.51 (1H, dt, *J* = 2; 12 Hz, CHCH_2_CH); 1.59–1.70 (2H, m, CHCH_2_CH_2_); 2.14 (3H, s, NHCOCH_3_); 2.23–2.35 (2H, m, CHCH_2_Ac); 2.36 (3H, s, Ar-Me); 3.28–3.43 (2H, m, CH_2_NH); 3.85–3.89 (1H, m, CH); 4.18–4.23 (1H, m, CH); 7.32 (1H, d, *J* = 8 Hz, H4); 7.42 (1H, s, H6); 7.68 (1H, d, *J* = 8 Hz, H3). ^13^C NMR (100 MHz, CD_3_OD) δ: 19.88; 21.02; 23.87; 28.40; 30.40; 36.13; 37.32; 37.47; 43.65; 67.61; 68.49; 81.82; 100.08; 115.79 (t, *J* = 252 Hz); 126.95 (t, *J* = 24 Hz); 127.33 (t, *J* = 9 Hz); 127.58; 133.23; 136.75 (t, *J* = 4 Hz); 166.65 (t, *J* = 30 Hz); 171.73; 172.07. ^19^F (376 MHz, CD_3_OD) δ: −104.58. IR (KBr, cm^−1^): 3348; 3246; 2985; 2935; 2864; 1728; 1683; 1666; 1519; 1367; 1267; 1151; 1083.

*tert*-Butyl 2-((4*R*,6*R*)-6-(2-(2-(2-acetamido-5-chlorophenyl)-2,2-difluoroacetamido)ethyl)-2,2-dimethyl-1,3-dioxan-4-yl)acetate (**13c**)

White solid. Yield: 58%. m.p. 128–130 °C. HRMS (ESI) calc. for C_24_H_33_ClF_2_N_2_O_6_ 518.1995, found [M + 1]^+^ 518.1999. HPLC grade: 96%. ^1^H NMR (400 MHz, CD_3_OD) δ: 1.13 (1H, q, *J* = 12 Hz, CHCH_2_CH); 1.26 (3H, s, C(CH_3_)_2_); 1.35 (3H, s, C(CH_3_)_2_); 1.44 (9H, s, C(CH_3_)_3_); 1.52 (1H, dt, *J* = 2; 12 Hz, CHCH_2_CH); 1.60–1.70 (2H, m, CHCH_2_CH_2_); 2.15 (3H, s, NHCOCH_3_); 2.23–2.38 (2H, m, CHCH_2_Ac); 3.28–3.42 (2H, m, CH_2_NH); 3.86–3.90 (1H, m, CH); 4.19–4.23 (1H, m, CH); 7.52 (1H, dd, *J* = 2; 8 Hz, H4); 7.61 (1H, d, *J* = 2 Hz, H6); 7.89 (1H, d, *J* = 8 Hz, H3). ^13^C NMR (100 MHz, CD_3_OD) δ: 20.00; 24.07; 28.51; 30.52; 36.24; 37.45; 37.64; 43.78; 67.74; 68.53; 81.96; 100.20; 115.11 (t, *J* = 254 Hz); 127.16 (t, *J* = 9 Hz); 128.55 (d, *J* = 25 Hz); 128.84; 131.75; 132.81; 136.10 (t, *J* = 3 Hz); 166.16 (t, *J* = 32 Hz); 171.75; 172.20. ^19^F (376 MHz, CD_3_OD) δ: −104.84. IR (KBr, cm^−1^): 3344; 3240; 2980; 2937; 2868; 1735; 1683; 1666; 1516; 1367; 1263; 1153; 1089; 829.

*tert*-Butyl 2-((4*R*,6*R*)-6-(2-(2-(2-acetamido-5-bromophenyl)-2,2-difluoroacetamido)ethyl)-2,2-dimethyl-1,3-dioxan-4-yl)acetate (**13d**)

Pale brown solid. Yield: 25%. m.p. 126–128 °C. HRMS (ESI) calc. for C_24_H_33_BrF_2_N_2_O_6_ 562.1490, found [M + 1]^+^ 562.1472. HPLC grade: 83%. ^1^H NMR (400 MHz, acetone-d_6_) δ: 1.13 (1H, q, *J* = 12 Hz, CHCH_2_CH); 1.27 (3H, s, C(CH_3_)_2_); 1.37 (3H, s, C(CH_3_)_2_); 1.43 (9H, s, C(CH_3_)_3_); 1.56 (1H, dt, *J* = 2; 12 Hz, CHCH_2_CH); 1.62–1.78 (2H, m, CHCH_2_CH_2_); 2.13 (3H, s, NHCOCH_3_); 2.22–2.41 (2H, m, CHCH_2_Ac); 3.34–3.53 (2H, m, CH_2_NH); 3.94–4.01 (1H, m, CH); 4.20–4.28 (1H, m, CH); 7.67 (1H, s, H4); 7.68 (1H, s, H6); 8.20 (1H, d, *J* = 8 Hz, H3). ^13^C NMR (100 MHz, acetone-d_6_) δ: 19.86; 24.50; 28.28; 30.42; 35.68; 36.97; 37.56; 43.33; 67.05; 68.10; 80.47; 99.28; 114.66 (t, *J* = 255 Hz); 118.01; 125.79 (t, *J* = 25 Hz); 126.53; 129.08 (t, *J* = 9 Hz); 135.33; 137.40 (t, *J* = 3 Hz); 165.36 (t, *J* = 30 Hz); 168.70; 170.35. ^19^F (376 MHz, acetone-d_6_) δ: -104.82. IR (KBr, cm^−1^): 3356; 2989; 2930; 1735; 1685; 1668; 1517; 1367; 1263; 1155; 1091.

#### 4.2.3. General Preparation of Sodium (3*R*,5*R*)-7-(2-(2-Acetamido-5-substituted-phenyl)-2,2-difluoroacetamide)-3,5-dihydroxyheptanoate salts **14** (**a**–**d**)

Product **13** (**a**–**d**) (1–3 mmol) and methanol (5 mL) were added under magnetic stirring at room temperature. After 30 min, was added 0.5 mL of 4% HCl solution (*w*/*v*) after some hours, the pH was measured (pH 1–2), and aqueous NaOH solution 7% (*w*/*v*) was added until pH 11–12 again. The stirring was switched off when the pH was measured at pH 7 and was added 5 mL of distilled MeOH/H_2_O (3:7) and activated carbon, which was under magnetic stirring for 5 min. The solution was evaporated and the products were precipitated in acetone giving the salts **14** (**a**–**d**). Overall reaction time was 48 h.

Sodium (3*R*,5*R*)-7-(2-(2-acetamido-phenyl)-2,2-difluoroacetamide)-3,5-dihydroxyheptanoate salt (**14a**)

White solid. Yield: 84%. m.p. > 240 °C. HRMS (ESI) calc. for C_17_H_21_F_2_N_2_NaO_6_ 410.1265, found [M+1]^+^ 410.1273. HPLC grade: 86%. ^1^H NMR (400 MHz, D_2_O) δ: 1.61–1.84 (4H, m, CHCH_2_CH; CHCH_2_CH_2_); 2.17 (3H, s, COMe); 2.31–2.41 (2H, m, CHCH_2_Ac); 3.39 (2H, t, *J* = 7 Hz, CH_2_NH); 3.78–3.84 (1H, m, CH); 4.07–4.13 (1H, m, CH); 7.40 (1H, d, *J* = 8 Hz, H6); 7.54 (1H, t, *J* = 8 Hz, H4); 7.65 (1H, t, *J* = 8 Hz, H5); 7.79 (1H, d, *J* = 8 Hz, H3). ^13^C NMR (100 MHz, D_2_O) δ: 22.41; 35.21; 36.87; 43.12; 44.98; 67.45; 67.55; 114.29 (t, *J* = 251 Hz); 127.05 (t, *J* = 8 Hz); 128.44; 129.17; 130.07; 132.68; 133.67; 165.37 (t, *J* = 30 Hz); 174.32; 180.42. ^19^F (376 MHz, D_2_O) δ: -101.07. IR (in MeOH, cm^−1^): 3420; 2950; 2854; 1710; 1490; 1415; 1190; 1065.

Sodium (3*R*,5*R*)-7-(2-(2-acetamido-5-methylphenyl)-2,2-difluoroacetamide)-3,5-dihydroxyheptanoate salt (**14b**)

White solid. Yield: 33%. m.p. > 240 °C. HRMS (ESI) calc. for C_18_H_23_F_2_N_2_NaO_6_ 424.1422, found [M+1]^+^ 424.1415. HPLC grade: 91%. ^1^H NMR (400 MHz, MeOD) δ: 1.55–1.78 (4H, m, CHCH_2_CH; CHCH_2_CH_2_); 2.14 (3H, s, COMe); 2.27–2.37 (2H, m, CHCH_2_Ac); 2.38 (3H, s, ArMe); 3.37 (2H, t, *J* = 7 Hz, CH_2_NH); 3.67–3.82 (1H, m, CH); 4.05–4.12 (1H, m, CH); 7.32 (1H, d, *J* = 8 Hz, H4); 7.43 (1H, s, H6); 7.62 (1H, d, *J* = 8 Hz, H3). ^13^C NMR (100 MHz, MeOD) δ: 21.02; 23.79; 37.00; 37.17; 44.81; 45.50; 69.00; 69.15; 115.78 (t, *J* = 251 Hz); 119.3; 127.40 (t, *J* = 8 Hz); 127.89; 133.19; 134.18; 137.00; 166.75 (t, *J* = 7 Hz); 172.02; 180.34. ^19^F (376 MHz, MeOD) δ: -101.17. IR (KBr, cm^−1^): 3322 (broad); 2925; 1663; 1556; 1514; 1424; 1273; 1184; 1151; 1077.

Sodium (3*R*,5*R*)-7-(2-(2-acetamido-5-chlorophenyl)-2,2-difluoroacetamide)-3,5-dihydroxyheptanoate salt (**14c**)

White solid. Yield: 13%. m.p. > 240 °C. HRMS (ESI) calc. for C_17_H_20_ClF_2_N_2_NaO_6_ 444.0876, found [M+1]^+^ 444.0867. HPLC grade: 75%. ^1^H NMR (400 MHz, MeOD) δ: 1.58–1.76 (4H, m, CHCH_2_CH; CHCH_2_CH_2_); 2.15 (3H, s, COMe); 2.30–2.34 (2H, m, CHCH_2_Ac); 3.38 (2H, t, *J* = 7 Hz, CH_2_NH); 3.80 (1H, s, CH); 4.07 (1H, s, CH); 7.52 (1H, dd, *J* = 0,5; 2 Hz, H4); 7.62 (1H, d, *J* = 0,5 Hz, H6); 7.84 (1H, d, *J* = 2 Hz, H3). ^13^C NMR (100 MHz, MeOD) δ: 23.89; 37.04; 38.16; 44.83; 45.54; 69.20; 69.21; 115.00 (t, *J* = 253 Hz); 127.17 (t, *J* = 8 Hz); 128.50; 129.00; 131.81; 132.66; 135.90; 166.10; 171.87. ^19^F (376 MHz, MeOD) δ: -101.88. IR (KBr, cm^−1^): 3734–3649; 3325–3064; 2972; 2941; 2929; 1670; 1597; 1558; 1423; 1261; 1153; 1085.

Sodium (3*R*,5*R*)-7-(2-(2-acetamido-5-bromophenyl)-2,2-difluoroacetamide)-3,5-dihydroxyheptanoate salt (**14d**)

Yellow solid. Yield: 24%. m.p. > 240 °C. HRMS (ESI) calc. for C_17_H_20_BrF_2_N_2_NaO_6_ 488.0371, found [M+1]^+^ 488.0369. HPLC grade: 97%. ^1^H NMR (400 MHz, MeOD) δ: 1.56–1.78 (4H, m, CHCH_2_CH; CHCH_2_CH_2_); 2.15 (3H, s, COMe); 2.23–2.35 (2H, m, CHCH_2_Ac); 3.38 (2H, t, *J* = 7 Hz, CH_2_NH); 3.79–3.82 (1H, m, CH); 4.04–4.08 (1H, m, CH); 7.66 (1H, dd, *J* = 2; 9 Hz, H4); 7.75 (1H, d, *J* = 2 Hz, H6); 7.80 (1H, d, *J* = 9 Hz, H3). ^13^C NMR (100 MHz, MeOD) δ: 23.94; 37.00; 38.17; 44.85; 45.56; 69.05; 69.19; 114.91 (t, *J* = 253 Hz); 119.08; 128.89; 129.06; 130.06 (t, *J* = 9 Hz); 135.71; 136.41; 166.13 (t, *J* = 31 Hz); 171.79; 180.41. ^19^F (376 MHz, MeOD) δ: -101.58. IR (KBr, cm^−1^): 3326 (broad); 2940; 1663; 1556; 1507; 1404; 1261; 1153; 1076.

### 4.3. Biological Evaluation

#### 4.3.1. Bacteria

*Salmonella enterica* serovar *Typhimurium* (*S. typhimurium)* strains TA97, TA98, TA100, TA104 and TA102 from the authors’ stock were used as described in the mutagenicity assay [42].

#### 4.3.2. Cell Cultures

Human hepatocellular carcinoma (HepG2) and Chinese Hamster Ovary (CHO-K1) cells obtained from the American Type Culture Collection (Manassas, VA, USA) were cultured in Eagle’s medium (MEM, Gibco^®^, Waltham, MA, USA) containing 10% fetal bovine serum (FBS) plus 100 µg/mL streptomycin and 100 µg/mL penicillin at 37 °C in a 5% CO_2_ atmosphere. Logarithmic-phase cells were used in all the experiments [48].

Mouse fibroblasts (L929) obtained from the American Type Culture Collection (Manassas, VA, USA) were cultured at 37 °C in RPMI-1640 medium (Gibco BRL) supplemented with 10% FBS and 2 mM glutamine, as reported in [49].

Primary cultures of cardiac cells (CC) were obtained as reported in [50]. The cultures were sustained in Dulbecco’s modified Eagle’s medium (DMEM) supplemented with 10% horse serum, 5% fetal bovine serum (FBS), 2.5 mM CaCl_2_, 1 mM L-glutamine, and 2% chicken embryo extract. Cell cultures were maintained at 37 °C in an atmosphere of 5% CO_2_, and assays were run at least three times in triplicate.

#### 4.3.3. Cellular Viability in Cell Cultures

Fresh HepG2 and CHO-K1 cells were seeded at a density of 1 × 10^4^/well. The water-soluble tetrazolium salt assay (WST-1) (4-[3-(4-iodophenyl)-2-(4-nitrophenyl)-2H-5-tetrazolio]-1,3-benzene disulfonate) (Roche Co., South San Francisco, CA) was used to determine the number of viable cells after 3 and 24 h of exposure to the compounds (0 to 1000 μM). Briefly, after treatment, the culture medium was replaced by 90 μL fresh culture medium and 10 μL WST-1 reagent and incubated at 37 °C and 5% CO_2_ for 3 h. The absorbance was then measured at 440 nm according to the kit protocol [51]. The intensity of the yellow color in the negative control (DMSO 1%) wells was designated as 100% viability and all further comparisons were based upon this reference level to determine the lethal concentration (LC_50_) to 50% of cultured cells.

Cardiac cell cultures were incubated for 24 h at 37 °C with different concentrations of the compounds (up to 200 µM) diluted in DMEM (without phenol red). Their morphology and spontaneous contractibility were evaluated by light microscopy and then the cellular viability was determined by the PrestoBlue^®^ assay. For this colorimetric bioassay, PrestoBlue^®^ (Invitrogen, Waltham, MA, USA) was added to each well (using a 10:1 medium/dye), the plate was further incubated for 24 h followed by optical density (OD) measurement performed at 570 and 600 nm, as recommended by the manufacturer. A similar protocol was used to evaluate L929 cell viability after 96 h of exposure to the compounds, using 10 μL AlamarBlue^®^ (Invitrogen). The results were expressed as the difference in the percentage of reduction between treated and untreated cells and the LC_50_ value, which corresponds to the concentration that reduces by 50% the cellular viability [52].

#### 4.3.4. Cultures of *P. falciparum*-Infected Erythrocytes and *In Vitro* Assays

The chloroquine-resistant and mefloquine-sensitive [53] *P. falciparum* W2 clone was maintained in continuous culture as previously described [54]. Briefly, parasites were cultivated in human erythrocytes (A+) at 37 °C in Petri dishes using complete medium (RPMI 1640 supplemented with 10% human sera blood group A+, 2% glutamine and 7.5% NaHCO_3_) and kept either in a candle jar, or in an environment containing a gas mixture atmosphere (3% O_2_, 5% CO_2_ and 91% N_2_). Prior to testing, the ring-stage parasites were synchronized using sorbitol [55]. The parasite suspension was adjusted for 0.05% parasitemia and 1.5% hematocrit and then distributed in 96-well microtiter plates (Corning, Santa Clara, CA, USA), 180 μL per well, to which 20 μL of different concentrations of the test drugs and controls had previously been added. The maximum concentration of 50 μg/mL (~157 μM) was tested at least three times for each compound against *P. falciparum*; the drug activity was evaluated using the anti-HRPII assay [56] with commercially available monoclonal antibodies (MPFM ICLLAB-55A^®^, MPFG55P ICLLAB^®^,Portland, OR, USA) raised against a *P. falciparum* histidine and alanine-rich protein (HRP2).

The test quantification was read at 450 nm on a spectrophotometer (SpectraMax340PC384, Molecular Devices Sunnyvale, CA, USA), and the drug activity was expressed as the half-maximal inhibitory concentration (EC_50_) compared to the drug-free controls using the curve-fitting software Origin 8.0 (OriginLab Corporation, Northampton, MA, USA) [57].

#### 4.3.5. Trypanocidal *In Vitro* Phenotypic Screening

Bloodstream trypomastigote forms (BT) of the Y strain were obtained from infected albino Swiss mice at the peak of parasitemia, isolated by differential centrifugation and resuspended in DMEM to a parasite concentration of 10^7^ cells/mL in the presence of 10% of mouse blood. This suspension (100 µL) was added in the same volume of **13** (**a**–**d**) and **14** (**a**–**d**) previously prepared at twice the desired final concentrations. Cell counts were performed in the Neubauer chamber and the trypanocidal activity was expressed as EC_50_, corresponding to the concentration that leads to the lysis of 50% of the parasites [57].

Tissue culture-derived trypomastigotes (Tulahuen strain expressing the *E. coli* β-galactosidase gene) were maintained in L929 cell lines and collected from the supernatant after 96 h of parasite infection, following previously established protocols [57]. Briefly, after 24 h of L929 platting (4 × 10^3^ cells/well), the cultures were incubated with trypomastigotes (10:1 parasite/host cell ratio) for 2 h at 37 °C. Then, the cell cultures were rinsed to remove non-internalized parasites and then further incubated for 24 h at 37 °C. Then the infected cultures were followed using two sequential protocols. In the first one, the infected cultures were exposed to each tested compound diluted at a fixed concentration of 10 µM (that corresponds to the EC_90_ value of Bz) in RPMI and incubated for 96 h at 37 °C. After this period, 50 µL chlorophenol red glycoside (500 µM) in 0.5% Nonidet P40 was added to each well and the plate was incubated for more than 18 h at 37 °C, after which the absorbance was measured at 570 nm. Then, the compounds were next evaluated using dose–response assays in which Tulahuen-infected-L929 cells were exposed to different non-toxic concentrations of each compound (0–50 µM) and then after 96 h of incubation the same procedure was conducted as above reported. Controls with untreated and Bz-treated cells are run in parallel. The results were expressed as the percent difference in the reduction between treated and untreated cells [57].

#### 4.3.6. *Salmonella*/Microsome Assay

The mutagenicity reverse mutation test was carried out to investigate the potential of **13** (**a**–**d**) to induce genetic mutation in *S. typhimurium* TA97, TA98, TA100, TA102 and TA104 strains. The test was carried out according to the pre-incubation method, both in the absence and presence of a metabolic activation system (4% S9 mix, Aroclor-pre-induced, from Moltox Inc., Boone, NC, USA). DMSO 10% served as the negative control while known mutagens were used as the positive control substances. The positive controls without the S9 mix were: 4-nitroquinoline 1-oxide (4-NQO, CAS Number 56-57-5) (5 μg per plate) for TA97 and TA98; sodium azide (SA, CAS Number 26628-22-8) (10 mg per plate) for TA100; mitomycin C (MMC, CAS Number 50-07-7) (1 μg per plate) for TA102 and methyl methanesulfonate (MMS, CAS Number 66-27-3) (200 μg per plate) for TA104. The positive controls with the S9 mix were: 2-amineanthracene (2-AA, CAS Number 613-13-8) (10.0 μg per plate) for TA97 and TA98; and benz(a)pyrene (BaP, CAS Number 50-32-8) (50.0 mg per plate) for TA100, TA102 and TA104. A dose-finding test was carried out with and without the metabolic activation system (S9 mix) for each tester strain. A total of six concentrations were set, from 0 to 1500.0 μM.

For the assays without metabolic activation, 0.5 mL of 0.1 mol/L sodium-phosphate buffer (pH 7.4) was added, and for the assays in the presence of metabolic activation, 0.5 mL S9 mix was mixed with 0.1 mL culture medium (2 × 10^8^ cells/mL) plus 0.1 mL of each compound solutions. The mixtures were incubated in a shaker at 37 °C (pre-incubation). After 30 min pre-incubation under light protection, the mixture was added to and mixed with 2 mL top agar containing 0.05 mmol/L L-histidine and D-biotin for the *S. typhimurium* strains. Each of these was then spread on a minimum glucose agar plate. After the top agar solidified, the plates were incubated at 37 °C for 60–72 h. Each tester strain was assayed in triplicate, and the number of revertant colonies was counted for each tester strain and treatment group [58]. The results were judged to be positive when the average number of revertant colonies in each treated group increased with an increase in the compound concentration, reaching at least twice the number in the negative control group.

To determine the cytotoxic effects, after 30 min pre-incubation, the assay mixtures were diluted in 0.9% NaCl (*w*/*v*) to obtain a suspension containing 2 × 10^2^ cells/mL. A suitable aliquot (100 μL) of this suspension was plated on nutrient agar (0.8% bacto nutrient broth (Difco), 0.5% NaCl and 1.5% agar). The plates were then incubated at 37 °C for 24 h and the colonies counted. All the experiments were conducted in triplicate and were repeated at least twice. Statistical differences between the groups were analyzed by a two-way ANOVA (*p* < 0.05) and Tukey’s post hoc test.

#### 4.3.7. *In Vitro* Micronuclei Assay in the Cell Culture (MNvit)

Fresh HepG2 and CHO-K1 cells were seeded at the density of 1 × 10^5^ cells/mL into 24-well plates (1 mL/per well). The compounds **13** (**a**–**d**) were then added to the medium to final concentrations of 500, 1000 and 2000 µM and incubation continued for 3 h or 24 h. DMSO 10% was used as the negative control, and, BaP (80 µM) for HepG2 and MMC (5 µM) for CHO-K1 were the positive controls. After exposure to the compounds, the cells were incubated for more than 24 h under growth conditions before quantification of the micronuclei and cytotoxicity. The cytogenetic studies were carried out in triplicate as described previously (see [51]). In order to determine the mitotic index and the number of cells with micronuclei, the medium was replaced by a cold methanol-glacial acetic acid (3:1) fixative for 30 min, and the cells then rinsed with distilled water for 2 min and air dried. The fixed cells were stained with 4,6-diamidino-2-phenylindole (DAPI) (0.2 pg/mL) dissolved in McIlvaine buffer (0.1 M citric acid plus 0.2 M Na_2_HPO_4_, pH 7.0) for 60 min, washed with McIlvaine buffer for 5 min, briefly rinsed with distilled water and mounted in glycerol. To determine the mitotic index and the number of cells with micronuclei, 1000 cells per well (3000 cells per concentration) were analyzed under a fluorescence microscope. The results for micronuclei were presented as the percentage of cells containing micronuclei in the 3000 cells/concentration group analyzed. Cells that glowed brightly and had homogenous nuclei were considered as having normal phenotypic morphology. Apoptotic nuclei were identified by the condensed chromatin at the periphery of the nuclear membrane or by fragmented nuclear body morphology. Necrotic cells presented chromatin forms with irregularly shaped aggregates, a pyknotic nucleus (shrunken and darkly stained) and cell membrane disruption, with cellular debris spilled into the extracellular milieu. Once again, 3000 cells were counted and the percentage of viable cells was evaluated, discounting apoptotic and necrotic cells. Statistical differences between the groups were analyzed by two-way ANOVA (*p* < 0.01) and Tukey’s post hoc test.

## Figures and Tables

**Figure 1 pharmaceuticals-16-00782-f001:**
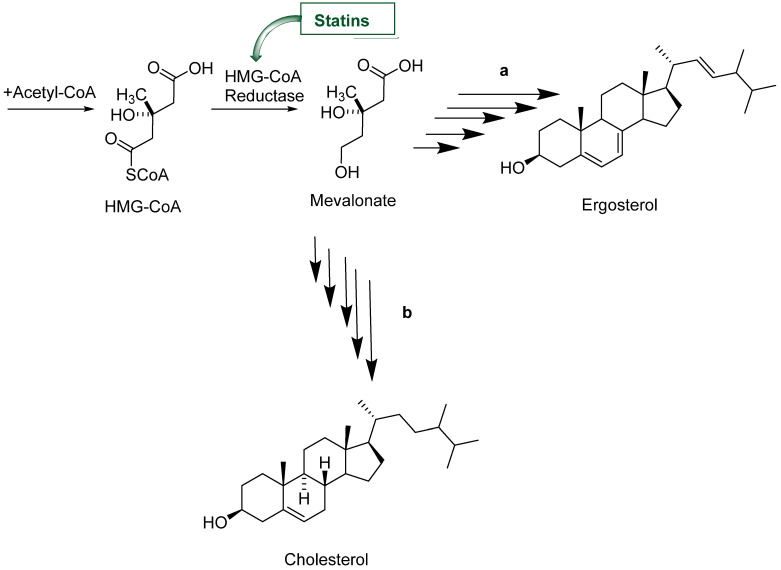
The simplified ergosterol (a) and cholesterol (b) biosynthesis pathway, highlighting the inhibition of hydroxy-3-methylglutaryl coenzyme A reductase (HMG-CoA) by statins. Ergosterol and cholesterol have a very conserved metabolic pathway and statins act inhibiting HMG-CoA reduction to mevalonate, which is a common substrate in both vias.

**Figure 2 pharmaceuticals-16-00782-f002:**
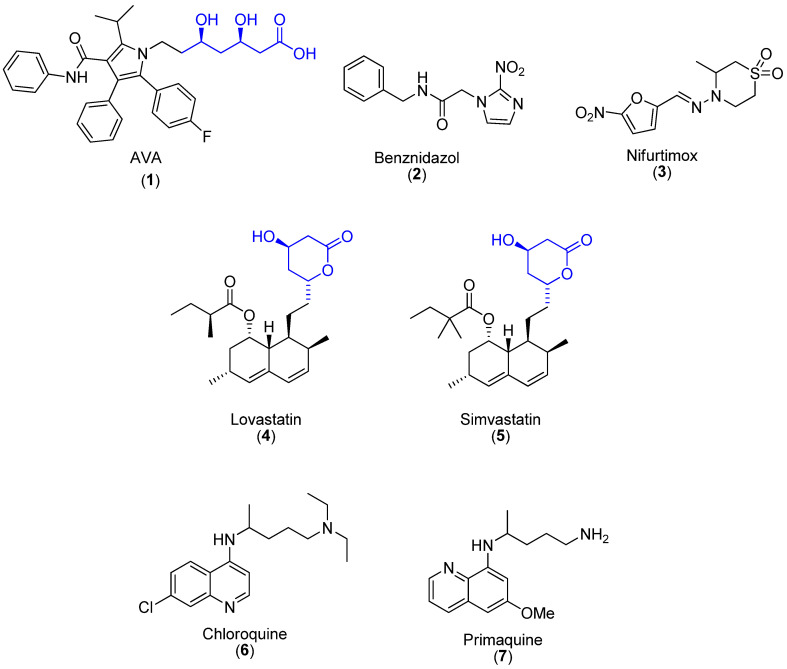
Chemical structure of antiparasitic compounds addressed in this study. (**1**) Atorvastatin (AVA) is a third-generation statin with antimicrobial potential. (**2**) Benznidazole is a nitroimidazolic drug, thriving as the first choice in Chagas Disease. (**3**) Nifurtimox is a nitrofuran outcasted in some countries mainly because of its adverse effects. (**4**) Lovastatin is a second-generation statin, which presents some antiprotozoan efficacy. (**5**) Simvastatin is a second-generation statin that presents some antiprotozoan efficacy. (**6**) Chloroquine is an aminoquinoline that is the first choice for the treatment of Malaria but could lead to several adverse effects. (**7**) Primaquine is an aminoquinoline with optimal efficacy in antiplasmodial therapy and fewer adverse effects.

**Figure 3 pharmaceuticals-16-00782-f003:**
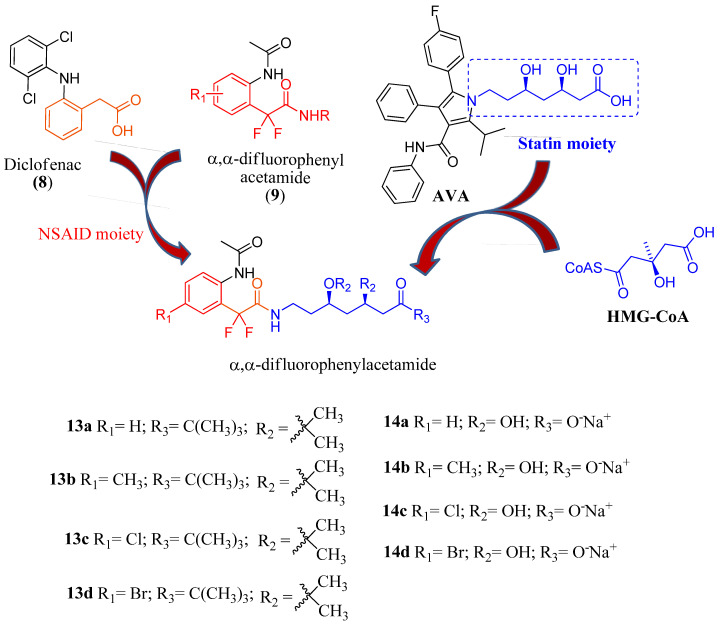
Synthetic design of novel hybrids compounds of α,α-difluorophenylacetamides with the moiety of statins (3*S*,5*S*)-3,5-dihydroxyheptanoic **13** (**a**–**d**) and **14** (**a**–**d**).

**Figure 4 pharmaceuticals-16-00782-f004:**
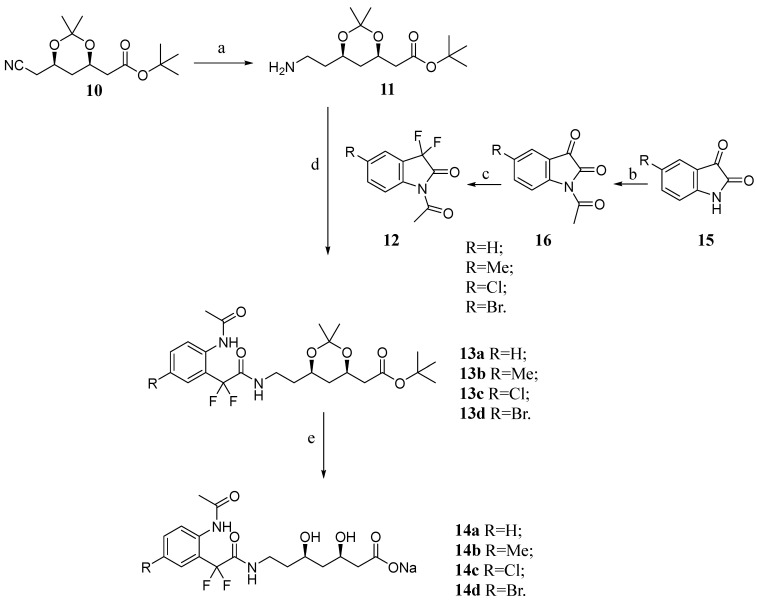
Synthesis of the novel atorvastatin analogs **13a**–**d** and **14a**–**d.** Experimental conditions: (a) H_2_ (50 psi), Ni-Ra or Pd (C), MeOH, NH4OH, rt, 48 h, 90%; (b) Acetic anhydride, 140 °C, 2–6 h, 82–89%; (c) DAST, CH_2_Cl_2_, rt, 18 h, 88–96%; (d) DMF, 70 °C, 24 h, 25–70%; (e) i. MeOH, HClaq 4% (*w*/*v*), rt; ii. NaOHaq.70% (*w*/*v*), rt, 48 h, 13–84**%**.

**Figure 5 pharmaceuticals-16-00782-f005:**
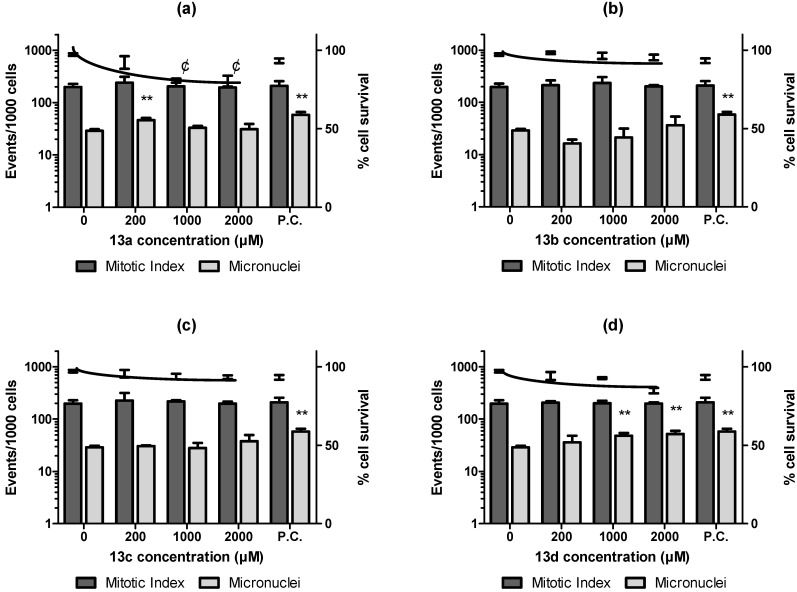
Mitotic index, micronuclei, and survival rate of CHO-K1 cells after 3 h of exposure with statin derivatives **13a**–**d**. Mitotic index, micronuclei, (left Y axis) and survival rate (right Y axis) of CHO-K1 cells. After 3 h of exposure, it is possible to observe that **13a** (**a**) was cytotoxic and induced micronuclei in CHO-K1 cells at 200 µM. There was no cytotoxic or genotoxic response of **13b** (**b**) and **13c** (**c**). **13d** (**d**) reduced the survival and induced micronuclei at 1000 and 2000 µM concentrations. ** *p* < 0.05 opposed to negative control in cytogenetics events. ¢ *p* < 0.05 vs. negative control in survival rates.

**Figure 6 pharmaceuticals-16-00782-f006:**
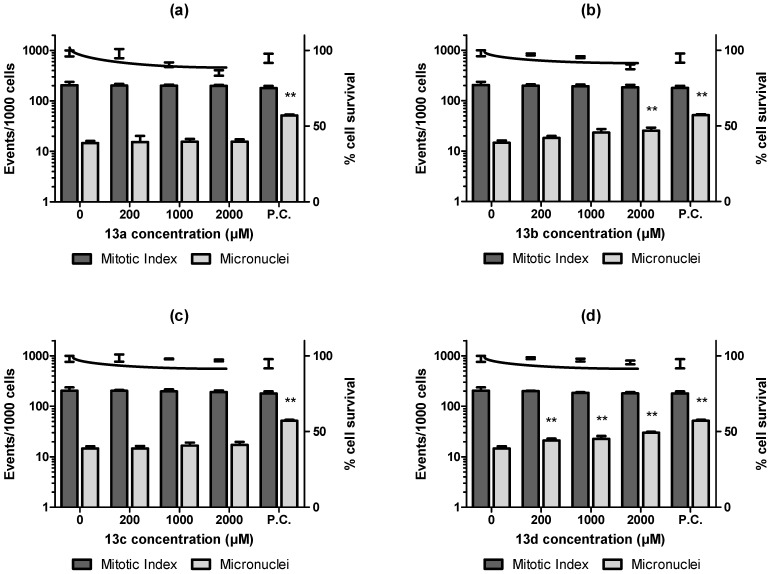
Mitotic index, micronuclei, and survival rate of HepG2 cells after 3 h of exposure with statin derivatives **13a**–**d**. Mitotic index, micronuclei (left Y axis) and survival rate (right Y axis) of HepG2 cells. After 3 h of exposure, it is possible to observe that **13a** (**a**) and **13c** (**c**) did not induce micronuclei formation in HepG2, but just **13a** presented cytotoxic effects upon 1000 µM. The compound **13b** (**b**) was genotoxic and cytotoxic to hepatic cells at 2000 µM and **13d** (**d**) it was non-cytotoxic and induced a significant micronuclei formation in HepG2 cells. ** *p* < 0.05 vs. negative control in cytogenetics events.

**Table 1 pharmaceuticals-16-00782-t001:** Evaluation of activity against *P. falciparum* parasites (W2 clone) chloroquine-resistant, cytotoxicity against a human hepatoma cell line (HepG2) and drug selectivity index (SI) of AVA, compounds **13** (**a**–**d**), **14** (**a**–**d**) and chloroquine as reference drug, after treatment for 24 h at 37 °C.

Compounds	EC_50_ (µM)	LC_50_ (µM)	SI
AVA	10.30 ± 1.2	>1000	>98
**13a**	>62	>1000	Inactive
**13b**	14.26 ± 0.2	>1000	>70
**13c**	11.78 ± 2.0	>1000	>85
**13d**	>89	>1000	Inactive
**14a**	>122	>488	Inactive
**14b**	>118	>472	Inactive
**14c**	>112	>450	Inactive
**14d**	>102	>409	Inactive
Chloroquine	0.59 ± 0.03	1219	2066

EC_50_: Efficacy concentration for 50% of parasite kill, evaluated in two to four different experiments for each test; LC_50_: lethal concentration for 50% of HepG2 cells; IS: selectivity index (LC_50_/EC_50_).

**Table 2 pharmaceuticals-16-00782-t002:** Evaluation of the activity against trypomastigote forms of *T. cruzi* (Y strain), cytotoxicity against mouse cardiac muscle cell cultures and drug selectivity index (SI) of AVA, compounds **13** (**a**–**d**), **14** (**a**–**d**) and Bz as reference drug, after treatment for 24 h at 37 °C.

Compounds	EC_50_ (µM)	LC_50_ (µM)	SI
AVA	7.07 ± 1.78	360.7 ± 18.2	51
**13a**	>500	>500	ND
**13b**	>500	>500	ND
**13c**	28.20 ± 0.75	>500	>17.7
**13d**	23.18 ± 3.13	>500	>21.6
**14a**	>500	>500	ND
**14b**	>500	>500	ND
**14c**	>500	>500	ND
**14d**	>500	>500	ND
Benznidazole	13.00 ± 2.0	>1000	>77

EC_50_: efficacy concentration for 50% of parasite kill, evaluated in two to four different experiments for each test; LC_50_: lethal concentration for 50% of murine cardiac cells; IS: selectivity index (LC_50_/EC_50_).

**Table 3 pharmaceuticals-16-00782-t003:** *In vitro* effect of the compounds against intracellular forms of *T. cruzi* (Tulahuen strain transfected with β-galactosidase), in the fixed concentration of 10 µM and corresponding EC_50_ (µM) values; besides cellular viability of L929 cultures exposed for 96 h to the studied compounds (LC_50_ values, µM) and their corresponding selectivity indexes (SI).

Compounds	10µM (% of Parasite Lysis)	EC_50_ (µM)	LC_50_ (µM)	SI
AVA	31.73 ± 5.80	45.33 ± 3.06	76.13 ± 2.16	1.7
**13a**	17.92 ± 11.02	>50	112.33 ± 28.29	ND
**13b**	25.41 ± 10.90	>50	113.67 ± 35.44	ND
**13c**	57.83 ± 2.84	9.24 ± 1.06	153.33 ± 25.17	16.7
**13d**	54.92 ± 16.94	11.42 ± 3.29	114.00 ± 5.29	10
Benznidazole	87.84 ± 7.08	1.83 ± 0.73	169.12 ± 27.2	92.4

EC_50_: Efficacy concentration for 50% of parasite kill, evaluated in two to four different experiments for each test; LC_50_: lethal concentration for 50% of L929 cells; IS: selectivity index (LC_50_/EC_50_).

**Table 4 pharmaceuticals-16-00782-t004:** Mean values (±SD) of revertant *His*^+^ colonies of *Salmonella enterica* serovar *Typhimurium* strains used in *Salmonella*/microsome assay after co-incubation with **13** (**a**–**d**).

		µM	TA97	TA98	TA100	TA102	TA104
**13a**	−S9	0	76 ± 6	25 ± 4	117 ± 5	386 ± 32	213 ± 3
	−S9	3	78 ± 6	24 ± 3	117 ± 43	458 ± 19	264 ± 27
	−S9	15	82 ± 8	29 ± 6	136 ± 13	435 ± 22	307 ± 18
	−S9	30	79 ± 8	31 ± 8	139 ± 13	433 ± 7	314 ± 26
	−S9	150	69 ± 10	32 ± 3	142 ± 5	380 ± 12	315 ± 36
	−S9	300	80 ± 11	**52 ± 4 ***	162 ± 37	345 ± 48	316 ± 33
	−S9	1500	90 ± 9	**61 ± 5 ***	179 ± 34	Cytotoxic	350 ± 21
	+S9	0	225 ± 9	29 ± 8	213 ± 20	344 ± 38	369 ± 44
	+S9	3	214 ± 21	25 ± 8	211 ± 37	384 ± 32	390 ± 49
	+S9	15	217 ± 9	25 ± 4	223 ± 15	399 ± 32	417 ± 15
	+S9	30	235 ± 23	26 ± 5	241 ± 36	407 ± 21	467 ± 19
	+S9	150	237 ± 26	27 ± 1	244 ± 14	411 ± 46	330 ± 9
	+S9	300	237 ± 4	33 ± 1	270 ± 45	448 ± 52	268 ± 37
	+S9	1500	290 ± 28	36 ± 3	290 ± 31	501 ± 20	Cytotoxic
**13b**	−S9	0	76 ± 6	15 ± 3	111 ± 13	370 ± 20	210 ± 3
	−S9	3	69 ± 12	18 ± 2	145 ± 11	388 ± 20	210 ± 11
	−S9	15	76 ± 7	19 ± 3	146 ± 1	393 ± 18	267 ± 15
	−S9	30	75 ± 12	21 ± 5	145 ± 17	423 ± 11	27 ± 13
	−S9	150	74 ± 12	22 ± 2	143 ± 40	503 ± 23	326 ± 13
	−S9	300	84 ± 7	22 ± 2	144 ± 8	543 ± 39	350 ± 22
	−S9	1500	89 ± 27	26 ± 4	178 ± 10	560 ± 41	400 ± 11
	+S9	0	185 ± 8	27 ± 8	183 ± 20	354 ± 38	314 ± 5,7
	+S9	3	189 ± 43	27 ± 5	179 ± 38	365 ± 37	355 ± 12
	+S9	15	183 ± 43	27 ± 1	188 ± 23	386 ± 42	356 ± 14
	+S9	30	216 ± 47	27 ± 5	228 ± 46	403 ± 19	421 ± 67
	+S9	150	231 ± 5	27 ± 5	260 ± 32	396 ± 31	447 ± 45
	+S9	300	244 ± 23	39 ± 1	306 ± 60	372 ± 40	449 ± 18
	+S9	1500	288 ± 33	45 ± 4	333 ± 45	399 ± 18	510 ± 13
**13c**	−S9	0	68 ± 5	21 ± 4	127 ± 5	385 ± 31	180 ± 9
	−S9	3	81 ± 6	21 ± 2	135 ± 38	373 ± 50	195 ± 9
	−S9	15	84 ± 8	21 ± 5	135 ± 21	386 ± 35	199 ± 7
	−S9	30	84 ± 6	21 ± 3	135 ± 30	387 ± 33	192 ± 3
	−S9	150	70 ± 6	21 ± 2	140 ± 31	386 ± 5	201 ± 17
	−S9	300	71 ± 11	21 ± 1	164 ± 17	400 ± 24	206 ± 8
	−S9	1500	62 ± 9	23 ± 4	168 ± 15	450 ± 32	241 ± 10
	+S9	0	225 ± 8	30 ± 4	213 ± 19	344 ± 19	369 ± 44
	+S9	3	219 ± 16	34 ± 4	223 ± 39	392 ± 8	384 ± 34
	+S9	15	245 ± 22	35 ± 3	223 ± 47	390 ± 12	401 ± 31
	+S9	30	247 ± 14	39 ± 3	228 ± 17	402 ± 24	445 ± 49
	+S9	150	249 ± 17	37 ± 7	278 ± 5	401 ± 66	305 ± 7
	+S9	300	259 ± 1	47 ± 4	286 ± 18	403 ± 12	297 ± 4
	+S9	1500	261 ± 13	49 ± 5	300 ± 15	430 ± 14	Cytotoxic
**13d**	−S9	0	71 ± 4	20 ± 3	103 ± 15	325 ± 2	403 ± 1
	−S9	3	119 ± 8	22 ± 7	99 ± 4	356 ± 7	469 ± 65
	−S9	15	137 ± 1	23 ± 5	96 ± 12	Cytotoxic	515 ± 61
	−S9	30	135 ± 4	25 ± 1	93 ± 1	-	570 ± 53
	−S9	150	134 ± 4	25 ± 1	Cytotoxic	-	605 ± 61
	−S9	300	**146 ± 6 ***	27 ± 5	-	-	615 ± 15
	−S9	1500	Cytotoxic	Cytotoxic	-	-	620 ± 31
	+S9	0	119 ± 3	24 ± 3	139 ± 17	124 ± 1	382 ± 30
	+S9	3	134 ± 3	28 ± 1	156 ± 18	153 ± 13	490 ± 52
	+S9	15	135 ± 1	28 ± 5	166 ± 11	155 ± 20	513 ± 19
	+S9	30	201 ± 4	31 ± 7	173 ± 1	160 ± 11	530 ± 7
	+S9	150	203 ± 1	34 ± 2	174 ± 21	162 ± 31	541 ± 29
	+S9	300	**288 ± 6 ***	34 ± 4	181 ± 15	176 ± 11	589 ± 5
	+S9	1500	**410 ± 10 ***	35 ± 3	Cytotoxic	Cytotoxic	591 ± 11

SD: standard deviation; −S9: absence of metabolic activation; +S9: presence of metabolic activation. Positive controls without S9: 4NQO (1.0 μg/pl.) for TA97, 286 ± 17 revertants and TA98 120 ± 10 revertants; AS (1.0 μg/pl.) for TA100, 607 ± 56 revertants; MMC (0.5 μg/pl.) for TA102, 2968 ± 34 revertants; MMS (50 µg/pl.) for TA104 746 ± 58 revertants. With S9: 2AA (1.0 μg/pl.) for TA97, 587 ± 11 revertants, for TA98, 305 ± 1 revertants and for TA100, 1436 ± 40 revertants; B[a]P (50 µg/pl.) for TA102, 1448 ± 79 revertants and for TA104 763 ± 11 revertants. Bold and * marked results indicate difference from the negative control in one-way ANOVA followed by Tukey’s post hoc test (*p* < 0.05).

**Table 5 pharmaceuticals-16-00782-t005:** Summarized outcomes of the activity of tested compounds against *P. falciparum* and *T. cruzi* and their corresponding safety profile concerning genotoxicity.

	Antiplasmodial Activity	Trypanocidal Activity	Bacterial Mutagenicity	Mammalian Cell Genotoxicity
**13a**	No	No	Yes	Yes
**13b**	Yes	No	No	Yes
**13c**	Yes	Yes	No	No
**13d**	No	Yes	Yes	Yes
**14a**	No	No	Not Analyzed	Not Analyzed
**14b**	No	No	Not Analyzed	Not Analyzed
**14c**	No	No	Not Analyzed	Not Analyzed
**14d**	No	No	Not Analyzed	Not Analyzed

## Data Availability

The datasets generated during and/or analyzed during the current study are available from the corresponding author on reasonable request.

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
