# Peer review of "Antiplasmodial, Trypanocidal, and Genotoxicity *In Vitro* Assessment of New Hybrid α,α-Difluorophenylacetamide-statin Derivatives"

_pharmaceuticals, 2023, doi:10.3390/ph16060782_

Round 1

Reviewer 1 Report

In the current preliminary study, the authors reported eight new hybrid compounds of α,α-difluorophenylacetam ides with the moiety of statins and assess their phenotypic activity against in vitro models of Plasmodium falciparum and Trypanosoma cruzi infection. 

Minor Comments: 

Line 28: "The" should be deleted. 

Line 29: Change to "a strategy" 

Line 31: As you know, there are two main types of intracellular parasites: Facultative and Obligate. Therefore, it should be mentioned in the text. 

Line 37: both parasite. Change to both "Parasites"

Line 42-43: All keywords should be provided according to MeSH terms at: http://www.nlm.nih.gov/mesh/MBrowser.html

Line 48: I recommended that all of the abbreviations should be complete at first use in the text and then use the abbreviation. E.g. Disability-adjusted life year (DALY). Please check them in the Text.

Line 57-59: Check this sentence. It is not understandable. 

Line 89: in 70 % of. Change to "70%" no need for space!% 

Line 191: Please check all subscripts according to the first one. E.g. LC50, EC50, Table 3. 

Line 276: VS Change to against or opposed to

Lines 276, 277, 285: P-value should be written "P" capital and Italic.

Line 496: Use one space between a number and its unit. Check it in the Text.

Line 502: Please check your unit. The unit degree centigrade "the degree (o)" should be superscript. 

Major Comments:

1) The introduction section is too long and does not have important points for the reader. Please summarize them. 

2) Line 76-77: Could you inform me about the aim of this sentence?! 

3) Line 518-521:Please check DO?! I think it should be changed to OD?! 

4) Line 634-635: Your statistical study is not clear!! 

5) Please improve your Materials and Methods section, especially the cell culture technique. For more your guide I suggested the following article in the text: 

DOI: 10.1016/j.micpath.2020.104438

6) Discussion should be comparable and written to the previous studies. I recommend the authors must use new reports of research articles after the years 2017 to 2023. Maximum 4-6 articles for comparing your results study. 

*) I recommend preparing a graphical abstract to improve the understanding of your study. 

**) Please write the research highlights according to your study. 

***) All changes in the manuscript should be identified in red font and yellow highlight.

I referred to some of them in my comments section. 

Author Response

Dear reviewer,

Thank you very much for the precious analyzes that certainly improved the quality of our manuscript.

Below you will find the responses to the reviewers.

Authors responses are highlighted in yellow.

Sincerely,

Review Report – Reviewer #1

Comments and Suggestions for Authors

In the current preliminary study, the authors reported eight new hybrid compounds of α,α-difluorophenylacetam ides with the moiety of statins and assess their phenotypic activity against in vitro models of Plasmodium falciparum and Trypanosoma cruzi infection. 

Answer: Thank you for your time in reviewing our manuscript.

Minor Comments: 

Line 28: "The" should be deleted. 

Answer: Done.

Line 29: Change to "a strategy" 

Answer: Done.

Line 31: As you know, there are two main types of intracellular parasites: Facultative and Obligate. Therefore, it should be mentioned in the text. 

Answer: Done.

Line 37: both parasite. Change to both "Parasites"

Answer: In this case, the phrase was “effect against both parasite forms” and then, we disagree in changing the sentence.

Line 42-43: All keywords should be provided according to MeSH terms at: http://www.nlm.nih.gov/mesh/MBrowser.html

Answer: According to the journal rules “Keywords: Three to ten pertinent keywords need to be added after the abstract. We recommend that the keywords are specific to the article, yet reasonably common within the subject discipline.” In this case, considering that MeSH terms are related to medical subjects and the manuscript also involves other areas as organic chemistry, we preferred to select more embracing keywords.

Line 48: I recommended that all of the abbreviations should be complete at first use in the text and then use the abbreviation. E.g. Disability-adjusted life year (DALY). Please check them in the Text.

Answer: Thank you for the recommendation. We adjusted the text as requested.

Line 57-59: Check this sentence. It is not understandable. 

Answer: Thank you, we rewrote the sentence, as your request. The new sentence is: “One of the most accepted pharmaceutical innovation strategies to accelerate the drug discovery pipeline is the repositioning of drugs, as the structure of a drug can interact with different pharmacophoric targets.”

Line 89: in 70 % of. Change to "70%" no need for space!% 

Answer: Thank you. Done

Line 191: Please check all subscripts according to the first one. E.g. LC50, EC50, Table 3. 

Answer: Thank you. Done

Line 276: VS Change to against or opposed to

Answer: Thank you. Done

Lines 276, 277, 285: P-value should be written "P" capital and Italic.

Answer: Thank you. Done

Line 496: Use one space between a number and its unit. Check it in the Text.

Answer: Thank you. Done

Line 502: Please check your unit. The unit degree centigrade "the degree (o)" should be superscript. 

 Answer: Thank you. Done

Major Comments:

1) The introduction section is too long and does not have important points for the reader. Please summarize them. 

Answer: Thank you for the suggestion. We rewrote de introduction section in order to attend your request, reducing in 2 paragraphs the text and improving/updating the references.

2) Line 76-77: Could you inform me about the aim of this sentence?! 

Answer: We intended to allow the reader to the risks and major side effects of statins, which could lead to patients withdraw in many cases. This fact is the biggest motivator for the search for new effective chemical entities, which have the same pharmacophoric site, but which have structural protectors to avoid these negative effects.

3) Line 518-521:Please check DO?! I think it should be changed to OD?! 

Answer: Thank you, we corrected the term to attend your request.

4) Line 634-635: Your statistical study is not clear!! 

Answer: We reported the recommended statistical analysis models in OECD test guidelines to determine the genotoxic potential of chemical entities.

5) Please improve your Materials and Methods section, especially the cell culture technique. For more your guide I suggested the following article in the text: 

DOI: 10.1016/j.micpath.2020.104438

Answer: Thank you for your suggestion. however, our research group and historical experience in culture of primary murine cardiomyocyte cells (MNCS: https://orcid.org/0000-0003-0078-6106) and also in culture of cell lines for assessment of toxicity (IF: https://orcid.org/0000-0003-1677-197X) in accordance with protocols such as the OECD and the ATCC. The suggested reference does not fit the models used by us in this study, as it is a different type of pathogen and cell lineage different from those used for our investigations. Thanks for the suggestion, but we chose not to include the citation in our manuscript.

6) Discussion should be comparable and written to the previous studies. I recommend the authors must use new reports of research articles after the years 2017 to 2023. Maximum 4-6 articles for comparing your results study. 

Answer: We consider that the results and discussion section of this manuscript needed to contemplate multidisciplinary aspects, which allow the reader to understand the rationale for the synthesis of these new compounds, their effectiveness against different protozoa and their toxicity. A total of 18 studies were used to discuss 4 major lines of investigation (synthesis, antiplasmodial activity, trypanocidal activity and genotoxicity), resulting in approximately 4-6 articles for each theme. Aspects that may be related to the structure-activity relationship and structure-toxicity relationship of these compounds were also raised in the discussion, such as the role of the bromine substituent in inducing mutagenicity and the effectiveness of the chlorine substituent in the most bioactive compound with the best leader profile. We improved the section, including other references, a summarizing table (as requested by another reviewer) and revising the english typos along the text. The lack of further studies involving the subject resulted in the need to use older references, reinforcing the importance of publishing this article to make a relevant contribution to scientific knowledge about this subject.

*) I recommend preparing a graphical abstract to improve the understanding of your study. 

Answer: Thank you. We prepared a graphical abstract in order to attend your request.

**) Please write the research highlights according to your study. 

Answer: Thank you. Done.

***) All changes in the manuscript should be identified in red font and yellow highlight.

Answer: We yellow highlighted the text in modified parts. The “track changes” tool was activated and the inserted and deleted parts were marked in red.

Reviewer 2 Report

In this manuscript, Araújo Lima et al. synthesize, characterize, and test novel compounds with potential anti-parasitic effects against Chagas disease and malaria. Eight compounds were tested in cellular models of Plasmodium falciparum and Trypanosoma cruzi infections. Their genotoxicity was also assessed. The most promising hit was compound 13c, which has shown high selectivity against T. cruzi and negligible genotoxicity, opening new perspectives for the treatment of Chagas disease. This study encourages future research on novel drugs against these two severe tropical diseases.

Overall, the manuscript is informative and clearly written. Nevertheless, there are a few typos and some figure legends should be more detailed (please see below). Methods are described in detail, facilitating the reproducibility of the present work.

Minor issues and recommendations for the authors:

1. In the abstract conclusion (line 39), please delete the "however" word, which seems unnecessary.

2. For clarity, in the Introduction, please include the full names of "DALYs" and "DTU".

3. In Figure 1, please place the AVA compound structure in the third place and update the numbering of the compounds in this figure and in the main text.

4. Please write more detailed legends for figures 2, 3, 4, 5, and 6. In the legend of Figure 2, please indicate what "A" and "B" refer to. In Figure 4, please delete the experimental conditions (if they are not required to understand this figure) and mention the meanings of "a", "b", "c", "d", and "e" in the legend. For consistency with figure 2, "a", "b", "c", "d", and "e" should be in capital bold letters.

5. The materials and methods section should be placed before the results or after the conclusion. I suggest relocating that section.

6. Throughout the manuscript, please use italics in the species names, as well as in "in vitro" and in "in vivo".

7. Throughout the manuscript, please write "EC50" as "EC50".

8. In lines 203 and 564, please confirm whether "EC90" is "EC90" and not "EC50".

9. In line 221, please correct the typo in "Against the intracellular forms of (DTU VI, Tulahuen strain)".

10. In the legends of figures 5 and 6, for increased clarity, please replace "µM" with the compound identification followed by "concentration (µM)". Please also place the legend titles under the figures.

11. Please include a summary table with the identification (13a-d/14a-d), full names of the eight synthesized compounds, and the conclusions for each one. The conclusions could be: effective against P. falciparum (yes/no), effective against T. cruzi (yes/no), genotoxic (yes/no).

There are a few typos, mentioned in the previous section.

Author Response

Dear reviewer,

Thank you very much for the precious analyzes that certainly improved the quality of our manuscript.

Below you will find the responses to the reviewers.

Authors responses are highlighted in yellow.

Sincerely,

Review Report Form – Reviewer #2

Comments and Suggestions for Authors

In this manuscript, Araújo Lima et al. synthesize, characterize, and test novel compounds with potential anti-parasitic effects against Chagas disease and malaria. Eight compounds were tested in cellular models of Plasmodium falciparum and Trypanosoma cruzi infections. Their genotoxicity was also assessed. The most promising hit was compound 13c, which has shown high selectivity against T. cruzi and negligible genotoxicity, opening new perspectives for the treatment of Chagas disease. This study encourages future research on novel drugs against these two severe tropical diseases.

Overall, the manuscript is informative and clearly written. Nevertheless, there are a few typos and some figure legends should be more detailed (please see below). Methods are described in detail, facilitating the reproducibility of the present work.

Answer: Thank you for your time in reviewing our manuscript.

Minor issues and recommendations for the authors:

  1. In the abstract conclusion (line 39), please delete the "however" word, which seems unnecessary.

Answer: Thank you. Done.

  1. For clarity, in the Introduction, please include the full names of "DALYs" and "DTU".

Answer: Thank you. Done.

  1. In Figure 1, please place the AVA compound structure in the third place and update the numbering of the compounds in this figure and in the main text.

Answer: We have entirely reviewed the introduction section to attend another reviewer request and, then, we adjusted the order of chemical compounds and so, considering the logical order of the compounds, AVA was maintained as the first compound. Thank you.

  1. Please write more detailed legends for figures 2, 3, 4, 5, and 6. In the legend of Figure 2, please indicate what "A" and "B" refer to. In Figure 4, please delete the experimental conditions (if they are not required to understand this figure) and mention the meanings of "a", "b", "c", "d", and "e" in the legend. For consistency with figure 2, "a", "b", "c", "d", and "e" should be in capital bold letters.

Answer: Thank you. When we rewrote the introduction section, we changed the order of figures 1 and 2, and also edited the legends of all figures, in order to improve the reader comprehension in each figure element. We edited the figures 1 and 4 to standardize the reaction steps and also included their meanings in the legends.

  1. The materials and methods section should be placed before the results or after the conclusion. I suggest relocating that section.

Answer: As requested, we have placed the methods section after the conclusion section.

  1. Throughout the manuscript, please use italics in the species names, as well as in "in vitro" and in "in vivo".

Answer: Thank you. Done.

  1. Throughout the manuscript, please write "EC50" as "EC50".

Answer: Thank you. Done.

  1. In lines 203 and 564, please confirm whether "EC90" is "EC90" and not "EC50".

Answer: Yes, we confirm we are talking about the efficacy concentration which kills 90% of parasites (EC90)

  1. In line 221, please correct the typo in "Against the intracellular forms of (DTU VI, Tulahuen strain)".

Answer: Thank you. Done.

  1. In the legends of figures 5 and 6, for increased clarity, please replace "µM" with the compound identification followed by "concentration (µM)". Please also place the legend titles under the figures.

Answer: Thank you. Done.

  1. Please include a summary table with the identification (13a-d/14a-d), full names of the eight synthesized compounds, and the conclusions for each one. The conclusions could be: effective against P. falciparum (yes/no), effective against T. cruzi (yes/no), genotoxic (yes/no).

Answer: As your request, we included the table 5, summarizing the results.

Comments on the Quality of English Language

There are a few typos, mentioned in the previous section.

Answer: Thank you. We reviewed the entire manuscript to improve the manuscript English language quality.

Round 2

Reviewer 1 Report

Accept

Accept